# Causal mapping of psychological and occupational risk factors for suicidal ideation in psychiatric nurses using Bayesian networks: A multicenter cross-sectional study

Min Wang[1], Yushun Yan[1], Wanqiu Yang[2,3], Ruini He[4], Lingdan Zhao[4], Yikai Dou[1], Yuanmei Tao[1], Xiao Yang[1], Qingqing Xiang[5]*, Xiaohong Ma[1]*

1 Mental Health Center and Institute of Psychiatry, West China Hospital, Sichuan University, Chengdu, China, 2 School of Ethnology and Sociology, Yunnan University, Kunming, China, 3 School of Medicine, Yunnan University, Kunming, China, 4 Department of Clinical Psychology, Zigong Mental Health Center, Zigong, China, 5 Department of Psychiatry, Zigong Affiliated Hospital of Southwest Medical University, Zigong, China

* maxiaohong@scu.edu.cn (XM); 651533793@qq.com (QX)

## Abstract

Psychiatric nurses represent a high-stress occupational group that experiences elevated levels of suicidal ideation (SI), emphasizing the need for focused mental health interventions. The main purpose of this study was to examine the prevalence of SI among psychiatric nurses and to identify the psychological and occupational factors associated with SI. A total of 1,835 psychiatric nurses completed questionnaires on depressive symptoms (PHQ-9), SI, quality of work-related life (QWL), and burnout. Multivariate logistic regression and phenotypic network analyses were conducted to identify factors associated with SI and the potential pathways linking depressive symptoms, burnout, and QWL to SI. The results indicated that 11.33% of the participants had SI in the past two weeks. Multivariate logistic regression revealed that emotional exhaustion, depersonalization, personal accomplishment, stress at work, general well-being, and the home-work interface were significant predictors of SI. Network analysis further revealed that psychomotor changes, guilt, sad mood, low energy, and appetite changes were the symptoms most directly associated with SI. In addition, sad mood, general well-being, and work-home interface were linked to job and career satisfaction, whereas sad mood and low energy were associated with emotional exhaustion and SI. These findings contribute valuable large-scale evidence on the mental health challenges faced by psychiatric nurses and highlight the importance of addressing mood disturbances, energy loss, and work-related stress in SI prevention efforts for this vulnerable group.

**Data availability statement:** Data are available from the public data repository OpenICPSR at https://www.openicpsr.org/openicpsr/project/237382/version/V1/view.

**Funding:** This work was supported by the Ministry of Science and Technology of the People's Republic of China (grant number 2022ZD0211700), the Key Research and Development Project of Zigong Science and Technology Program (grant number 2023YLWS11), the 135 Project from West China Hospital of Sichuan University (grant numbers 2023HXFH006, 2023HXFH040), and the Postdoctoral Research Fund of West China Hospital, Sichuan University (grant number 2024HXBH135).

**Competing interests:** The authors declare no conflicts of interest.

## Introduction

Suicide remains a significant global concern, causing over 817,000 deaths annually and accounting for 1.49% of all deaths worldwide [1]. Among healthcare professionals, psychiatric nurses are particularly vulnerable, and have significantly higher suicide rates than the general population does [2,3]. Elevated levels of suicidal ideation (SI) in this group are largely attributed to the intense emotional demands and chronic job-related stress inherent in their profession [4,5], highlighting the urgent need for targeted mental health support and interventions.

SI, as a precursor to suicide behavior, makes early identification crucial for prevention [6]. For healthcare workers, risk factors such as mental health problems, occupational stress, and personal difficulties exacerbate vulnerability to SI [7]. Among these factors, burnout has drawn particular attention, given its strong association with both personal and professional consequences [8,9]. The emotionally demanding nature of healthcare work, alongside long working hours and frequent exposure to traumatic situations, further highlights the importance of understanding these risks, particularly for psychiatric nurses [2]. In this context, both burnout and quality of work-related life (QWL) stand out as critical dimensions of work-related mental health that directly shape vulnerability to SI.

Burnout syndrome is typically conceptualized as a maladaptive response to prolonged and cumulative occupational stress, characterized by emotional exhaustion (EE), depersonalization (DP), and a diminished sense of personal accomplishment (PA) [10]. Nurses experiencing burnout face significant personal consequences, including strained interpersonal relationships, depression, anxiety, and heightened SI [8,11]. For example, a study conducted in Taiwan identified high levels of burnout and perpetual work stress as major contributors to nurses' SI [8], whereas research in the United States reported that 5.5% of nurses experienced SI attributable to burnout [12]. In addition to individual outcomes, burnout also leads to professional consequences such as decreased care quality, lower patient satisfaction, reduced productivity, and a greater risk of errors [13]. These issues not only harm patients and institutions but also create feedback loops that aggravate nurses' psychological strain and further increase their SI risk.

The QWL, defined as a worker's satisfaction with their professional life [12] and shaped by the relationship between the individual and their work environment [14], is another crucial determinant of nurses' mental health. QWL reflects the extent to which nurses' physical, emotional, and social needs are met through their work experience while simultaneously enabling organizational effectiveness [15]. Poor QWL has been consistently associated with burnout, depressive symptoms, and reduced job satisfaction [16]. Studies focused on psychiatric nurses further underscore the psychosocial risk and workload pressure specific to this group, highlighting that variations in sociodemographic and work characteristics can influence their vulnerability to SI [17].

Despite the recognized importance of burnout, QWL, and depressive symptoms in predicting suicide risk among healthcare professionals, most previous studies have examined these factors in isolation. This fragmented approach obscures the complex

and dynamic interactions among them and limits the understanding of their combined impact on the SI. Addressing this gap requires an integrative approach that can capture both the complexity of symptom interactions and the potential causal pathways underlying them.

Network analysis provides such a framework. Unlike traditional methods that typically focus on linear relationships, network analysis visualizes and quantifies the relationships among how symptoms cluster and interact within a broader structure [18]. This approach has been particularly useful in mental health research, offering insights into the interplay of multiple symptoms and highlighting critical nodes—symptoms or variables that exert a disproportionate influence on the network and may represent potential targets for interventions [19,20]. Complementing network analysis, directed acyclic graphs (DAGs) within a Bayesian network framework enable researchers to examine conditional dependencies and infer potential causal directions among variables [21]. By moving beyond correlation, DAGs help clarify which factors are more likely to directly shape the SI and which act through indirect pathways [22]. This capacity to map causal chains provides a richer understanding of how burnout, QWL, and depressive symptoms jointly contribute to SI among psychiatric nurses.

The purpose of this study was to investigate the prevalence of SI among psychiatric nurses and to examine how depressive symptoms, burnout, and QWL interact to influence SI. We hypothesized that higher levels of burnout and depressive symptoms, coupled with lower QWL, would be associated with an increased risk of SI. To test these hypotheses, we applied multivariate regression to identify significant predictors and employed network analysis together with Bayesian causal models to capture the complex interrelationships among these factors. This integrative methodological approach extends previous research by moving beyond correlation-based analyses to infer potential causal pathways, thereby offering novel insights into central and bridge symptoms that may serve as effective intervention targets to reduce SI and improve the mental health of psychiatric nurses.

## Methods

### Study design, setting, and participants

This study employed a cross-sectional design to investigate the network structure of SI, depressive symptoms, burnout, and quality of work-related life among psychiatric nurses. Participants were recruited from 23 psychiatric hospitals and mental health care facilities across Sichuan Province, China, between October 2023 and November 2023.

Data collection was conducted via a digital platform developed by the research team, through which participants completed a survey.

The inclusion criteria for participants were registered nurses, aged 18–60 years who were currently employed in psychiatric institutions and had at least one year of experience in psychiatric clinical nursing. The participants were required to be aware of the study and to agree to participate voluntarily. Exclusion criteria: Nurses who had left for more than a month, student interns, those without a nursing license, or those who provided invalid responses (e.g., responses with an average time of less than one second per item, uniform answers across all items, or incomplete surveys) were excluded.

All participants were informed that by selecting the consent boxes on the first page of the survey, they were providing informed consent. The participants were also informed that they could withdraw from the survey at any time and that their responses would not be shared with their employers. Ethics approval for the study was obtained from the Zigong Affiliated Hospital of Southwest Medical University (No. 2023-08-01). This study adheres to the principles of the Helsinki Declaration.

## Measures

### Depressive symptoms and suicidal ideation

Depressive symptoms were assessed via the Patient Health Questionnaire-9 (PHQ-9), a widely validated tool for screening and evaluating the severity of depression over the past two weeks [23]. The PHQ-9 includes nine items based on the

DSM-IV criteria for major depressive disorder, with total scores ranging from 0 to 27. In this study, item 9 of the PHQ-9 was specifically used to assess SI [24]. The participants were classified into two groups on the basis of their responses to this item: those who reported no suicidal ideation ("not at all") and those who reported any level of SI ("several days," "more than half the days," or "nearly every day"). In this study, the PHQ-9 demonstrated high reliability, with a Cronbach's α of 0.881.

### Occupational burnout

Occupational burnout was measured via the Chinese version of the 22-item Maslach Burnout Inventory-Human Services Survey (MBI-HSS), which includes three subscales: emotional exhaustion (9 items), depersonalization (5 items), and reduced personal achievement (8 items) [25–27]. This version has good reliability, with a Cronbach's α of 0.85 [28]. The MBI-HSS items are rated on a frequency scale, with higher scores for emotional exhaustion and depersonalization and lower scores for personal achievement indicating higher levels of burnout. In our study, the MBI-HSS demonstrated good reliability, with an overall Cronbach's α of 0.874. For the subscales, emotional exhaustion had an α of 0.835, depersonalization had an α of 0.724, and personal achievement had an α of 0.839, all of which are considered acceptable for reliability testing.

### Quality of work-related life

QWL was measured via the Chinese version of the Work-related Quality of Life Scale (WRQoL-2) [29]. In this study, the WRQoL-2 scale demonstrated strong reliability, with a Cronbach's α of 0.958. Among its subscales, working conditions (WCN) had an α of 0.893, stress at work (SAW) had an α of 0.780, control at work (CAW) had an α of 0.810, home-work interface (HWI) had an α of 0.770, employee engagement (EEN) had an α of 0.905, general well-being (GWB) was 0.859, and job and career satisfaction (JCS) had an α of 0.709.

### Statistical analysis

All analyses were performed via R version 4.4.0. Descriptive statistics were calculated for all variables to summarize the sample characteristics and distributions including SI, depressive symptoms, burnout, and QWL scores. All reverse-scored items were recoded to ensure consistency in direction. Continuous variables were described using means and standard deviations, medians with interquartile ranges (IQRs), and ranges (minimum-maximum), whereas categorical variables were summarized by frequency distributions and percentages. The participants were divided into two groups on the basis of the presence or absence of suicidal ideation (i.e., SI, nSI). Comparative analyses were conducted using t-tests for continuous variables and chi-square tests for categorical variables to examine differences in general clinical characteristics and symptoms between the two groups. Additionally, a multivariate logistic regression model was applied to identify factors associated with SI concerning burnout and QWL. A $p$-value < 0.05 was considered statistically significant.

### Network analysis

In this study, all symptoms were analyzed from a systems-based perspective. GGMs were employed to construct a symptom network via the qgraph package (version 1.9.2). In the network, nodes represent symptoms, and edges represent the partial correlation between two symptoms, adjusting for the effects of all the other variables in the network. The function qgraph allows visualization and analysis of the network by estimating partial correlations and applying LASSO regularization to simplify the network structure. GGMs assume that the data approximate multivariate normality; in this study, we examined distributional properties and further applied regularized estimation to mitigate potential violations of this assumption when estimating partial correlations among variables [18].

We employed the "flow" graphical function from the R package qgraph to identify symptoms directly associated with "SI" within the depression-burnout-QWL network. This function positions "SI" on the left and creates a vertical network to display its direct and indirect connections.

### Node centrality and bridge centrality indices

To identify key nodes within the network, strength, expected influence (EI), bridge strength, and bridge expected influence (bEI) were calculated via the R package networktools (version 1.5.0) according to previous study [30]. Strength is defined as the sum of the absolute values of the edges connected to a node. The EI defined as the sum of all edge weights connected to a node, considering both positive and negative associations. Nodes with higher centrality were more central and important in the network. Bridge strength is the sum of the absolute values of all edges connecting a node to nodes in other communities. The bEI measures the sum of the values of all the edges connecting a given node to all the nodes in the other communities. Higher bridging metric values suggest a greater likelihood of increasing the risk of contagion to other communities [31].

### Stability and accuracy

To assess the stability and accuracy of the network, we performed bootstrapping using the techniques implemented in the R package bootnet (version 1.6). To evaluate robustness, we conducted 1000 bootstrap samples for each estimation round, calculating confidence intervals for edge weights and centrality metrics. The stability of the node centrality and bridge centrality indices was measured using case-dropping bootstrapping, which calculates the correlation stability (CS) coefficient. This coefficient reflects the maximum proportion of cases that can be dropped while maintaining a 95% probability that the correlation between the original and subset centrality indices remains 0.7 or higher [32]. A CS coefficient above 0.75 indicates high stability [18].

### Directed acyclic graphs

A DAG is a model in Bayesian network that is used to explore the causal relationships between symptoms. DAGs provide information about the degree of conditional independence among variables and allow for causal inference on the basis of the data. The R package bnlearn (version 5.0.1) was utilized to construct the DAGs using the hill-climbing (hc) algorithm, which searches for the best network structure by optimizing a scoring criterion. To enhance the robustness and interpretability of the network, we additionally calculated the frequency of each edge across 10,000 bootstrap samples in the Bayesian network structures. This approach helps identify potential causal pathways between symptoms and other variables, offering insights into how changes in one symptom may lead to changes in others [21,22].

## Results

### Demographic and clinical status of the participants

Among the 2,017 participants initially recruited, 1,835 (90.98%) respondents provided fully qualified data on all three scales (Table 1). The sample was predominantly female (88.7%) and younger than 30 years (52.2%). More than half (54.6%) had attained a bachelor's degree or higher, and 68.1% were married. Most participants (73.6%) held junior or lower professional titles, whereas 29.7% reported a monthly income exceeding 5,000 RMB. Additionally, 39.5% had over 10 years of work experience, 46.3% engaged in night shifts more than four times per month, and 15.9% reported frequent overtime. Workplace violence was experienced by 28.6% of participants. Further details on burnout and the QWL can be found in S1 Table.

Among the 208 participants (11.33%) who reported experiencing SI, those with low income (12.33%), unmarried or divorced status (14.51%), and those experiencing workplace violence (14.5%) were more prevalent (all $p < 0.001$). Additionally, the scores of the burnout and QWL subdomains were significantly higher in the SI group than in the non-SI group ($p < 0.001$). These findings suggest that both personal and occupational stressors are likely contributors to SI and warrant further examination via multivariate analyses.

**Table 1. Characteristics of the participants.**

| Variable | Category | All sample (n = 1835) | Non-Suicidal Ideation (n = 1627) | Suicidal Ideation (n = 208) | Statistics | p value |
|---|---|---|---|---|---|---|
| Gender | | | | | 1.308a | 0.253 |
| | Male | 208 (11.34%) | 179 (86.06%) | 29 (13.94%) | | |
| | Female | 1627 (88.66%) | 1448 (89%) | 179 (11%) | | |
| Age (> 30years) | | | | | 1.725a | 0.189 |
| | No | 958 (52.21%) | 840 (87.68%) | 118 (12.32%) | | |
| | Yes | 877 (47.79%) | 787 (89.74%) | 90 (10.26%) | | |
| Education levels | | | | | 1.096a | 0.295 |
| | Below bachelor's degree | 833 (45.4%) | 731 (87.76%) | 102 (12.24%) | | |
| | Bachelor's degree and above | 1002 (54.6%) | 896 (89.42%) | 106 (10.58%) | | |
| Marry status | | | | | 8.151a | **0.004** |
| | Married | 1249 (68.07%) | 1126 (90.15%) | 123 (9.85%) | | |
| | Unmarried or others | 586 (31.93%) | 501 (85.49%) | 85 (14.51%) | | |
| Professional title | | | | | 2.447a | 0.118 |
| | Junior level and below | 1351 (73.62%) | 1188 (87.93%) | 163 (12.07%) | | |
| | Intermediate level and above | 484 (26.38%) | 439 (90.7%) | 45 (9.3%) | | |
| Income (>5000 RMB/month) | | | | | 3.914a | **0.048** |
| | No | 1290 (70.3%) | 1131 (87.67%) | 159 (12.33%) | | |
| | Yes | 545 (29.7%) | 496 (91.01%) | 49 (8.99%) | | |
| Work Experience (> 10 years) | | | | | 0.307a | 0.579 |
| | No | 1110 (60.49%) | 980 (88.29%) | 130 (11.71%) | | |
| | Yes | 725 (39.51%) | 647 (89.24%) | 78 (10.76%) | | |
| Night shift (> 4 times/month) | | | | | 0.232a | 0.63 |
| | No | 986 (53.73%) | 878 (89.05%) | 108 (10.95%) | | |
| | Yes | 849 (46.27%) | 749 (88.22%) | 100 (11.78%) | | |
| Overtime | | | | | 1.725a | 0.189 |
| | No | 1544 (84.14%) | 1376 (89.12%) | 168 (10.88%) | | |
| | Yes | 291 (15.86%) | 251 (86.25%) | 40 (13.75%) | | |
| Workplace force | | | | | 6.893a | **0.009** |
| | No | 1311 (71.44%) | 1179 (89.93%) | 132 (10.07%) | | |
| | Yes | 524 (28.56%) | 448 (85.5%) | 76 (14.5%) | | |
| Burnout | | | | | | |
| | Emotional exhaustion | 18.31 ± 10.34 | 17.26 ± 9.78 | 26.46 ± 10.99 | −11.497b | **<0.001** |
| | Personal achievement | 17.96 ± 10.7 | 17.25 ± 10.58 | 23.54 ± 10.06 | −8.443b | **<0.001** |
| | Depersonalization | 4.65 ± 5.24 | 4.14 ± 4.82 | 8.66 ± 6.5 | −9.691b | **<0.001** |
| Quality of work related-life | | | | | | |
| | Working conditions | 21.71 ± 4.05 | 22.02 ± 3.88 | 19.27 ± 4.51 | 8.39b | **<0.001** |
| | Stress at work | 19.95 ± 3.81 | 20.32 ± 3.67 | 17.05 ± 3.62 | 12.247b | **<0.001** |
| | Control at work | 18.74 ± 2.83 | 18.92 ± 2.75 | 17.27 ± 3 | 7.535b | **<0.001** |
| | Home-work interface | 7.61 ± 1.35 | 7.69 ± 1.31 | 7.02 ± 1.5 | 6.104b | **<0.001** |
| | Employee engagement | 18.42 ± 3.48 | 18.7 ± 3.3 | 16.25 ± 4.05 | 8.386b | **<0.001** |
| | General well-being | 18.45 ± 3.18 | 18.78 ± 2.98 | 15.88 ± 3.52 | 11.382b | **<0.001** |
| | Job and career satisfaction | 15.1 ± 2.09 | 15.25 ± 2 | 13.99 ± 2.43 | 7.162b | **<0.001** |

a = $\chi^2$, b = T; bold: Statistically significant differences.

## Factors associated with SI: results of multivariate logistic regression analyses

The multivariate logistic regression model (Fig 1, S2 Table) confirmed that several burnout and AWL variables were independently associated with SI. Emotional exhaustion (OR = 1.029, 95% CI: 1.007–1.05, $p = 0.011$), depersonalization (OR = 1.039, 95% CI: 1.004–1.075, $p = 0.027$), and personal accomplishment (OR = 1.029, 95% CI: 1.011–1.048, $p = 0.002$) were identified as significant risk factors. In contrast, lower stress at work (OR = 0.881, 95% CI: 0.83–0.934, $p < 0.001$), and greater general well-being (OR = 0.824, 95% CI: 0.757–0.896, $p < 0.001$) were protective. Additionally, home-work interfaces (OR = 1.264, 95% CI: 1.079–1.480, $p = 0.004$) played a notable role. While the effect sizes were generally modest, their cumulative impact suggests that burnout and poor QWL, when occurring simultaneously, substantially elevate the risk of SI. This highlights the importance of workplace interventions that target multiple dimensions rather than focusing on isolated factors.

## Phenotype network of burnout, QWL, depressive symptoms, and SI

The network model incorporated 19 psychological phenotypes, yielding 171 potential edges, of which 106 (62.0%) were retained after regularization (Fig 2A-C). Among these, 74 represented positive associations and 32 represented negative associations. The strongest direct edges connecting SI were with psychomotor changes, guilt, sad mood, energy, and appetite.

Centrality analysis revealed that low energy and emotional exhaustion as the most influential nodes in terms of strength, whereas sad mood was second-ranked in terms of expected influence (Fig 2D). Bridge analyses further revealed emotional exhaustion and low energy as critical connections linking occupational burnout and depressive symptoms (Fig 2E). Taken together, these results indicate that fatigue-related and affective symptoms not only cluster around SI but also act as bridges across symptom domains, underscoring their potential as key intervention targets.

## Bayesian network analysis

Bayesian network analysis provided complementary insights into the directionality of associations (Fig 3). General well-being influences several QWL domains, including employee engagement, work-related stress, control at work,

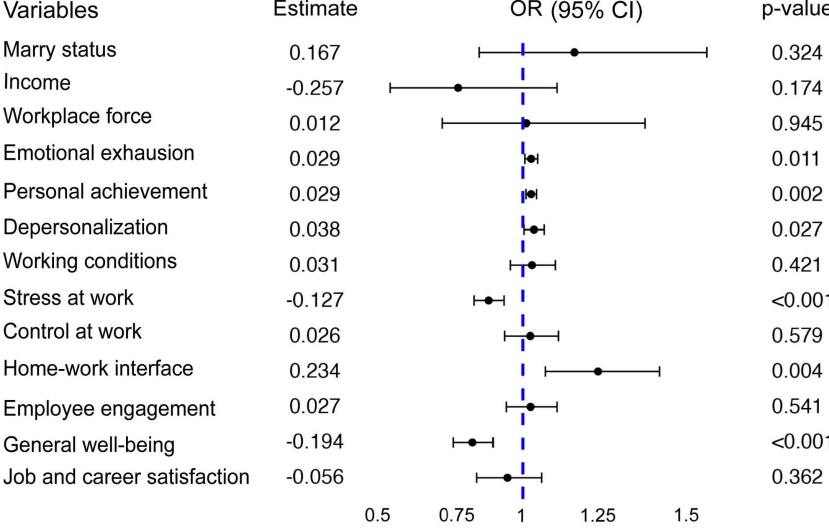

**Fig 1. Multivariate logistic regression analysis of predictors of suicidal ideation.** CI*: Confidence interval.

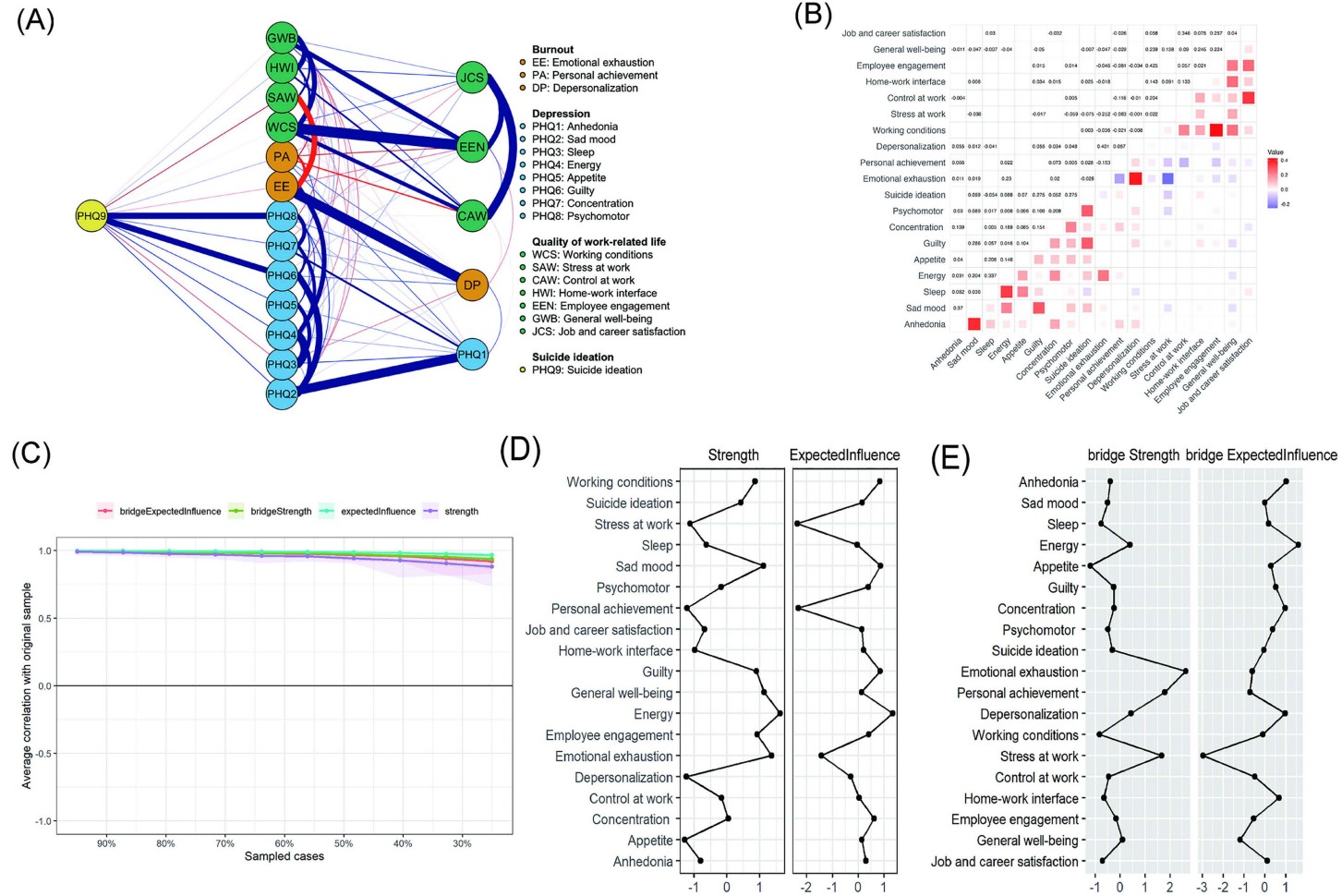

**Fig 2. The structural network of suicidal ideation, depression, burnout, and quality of work-related life among psychiatric nurses. (A)** Gaussian graphical model; **(B)** Edge weight matrix of symptom network; **(C)** Stability verification of central and bridging symptoms; **(D-E)** Central and bridge symptoms in the symptom network.

and the home-work interface, while also indirectly affecting burnout and depressive symptoms. Job and career satisfaction emerged as the most influential aspects of QWL, shaped by both occupational and emotional factors such as, control at work, working conditions, sleep disturbances and concentration issues. Among the depressive symptoms, sad mood was the most impactful and was associated with psychomotor changes, anhedonia, sleep disturbances, feelings of guilt, low energy, concentration difficulties, SI, depersonalization, emotional exhaustion, general well-being, and work-related stress. Psychomotor changes are driven primarily by other depressive symptoms but are also connected to depersonalization, highlighting the overlap between the affective and occupational domains. Finally, SI was most strongly predicted by sad mood, low energy, guilt, and psychomotor changes, which is consistent with the network findings. These patterns suggest that addressing both workplace-related and depressive factors, particularly those involving energy and mood regulation, may provide the most effective strategy for reducing SI risk among psychiatric nurses.

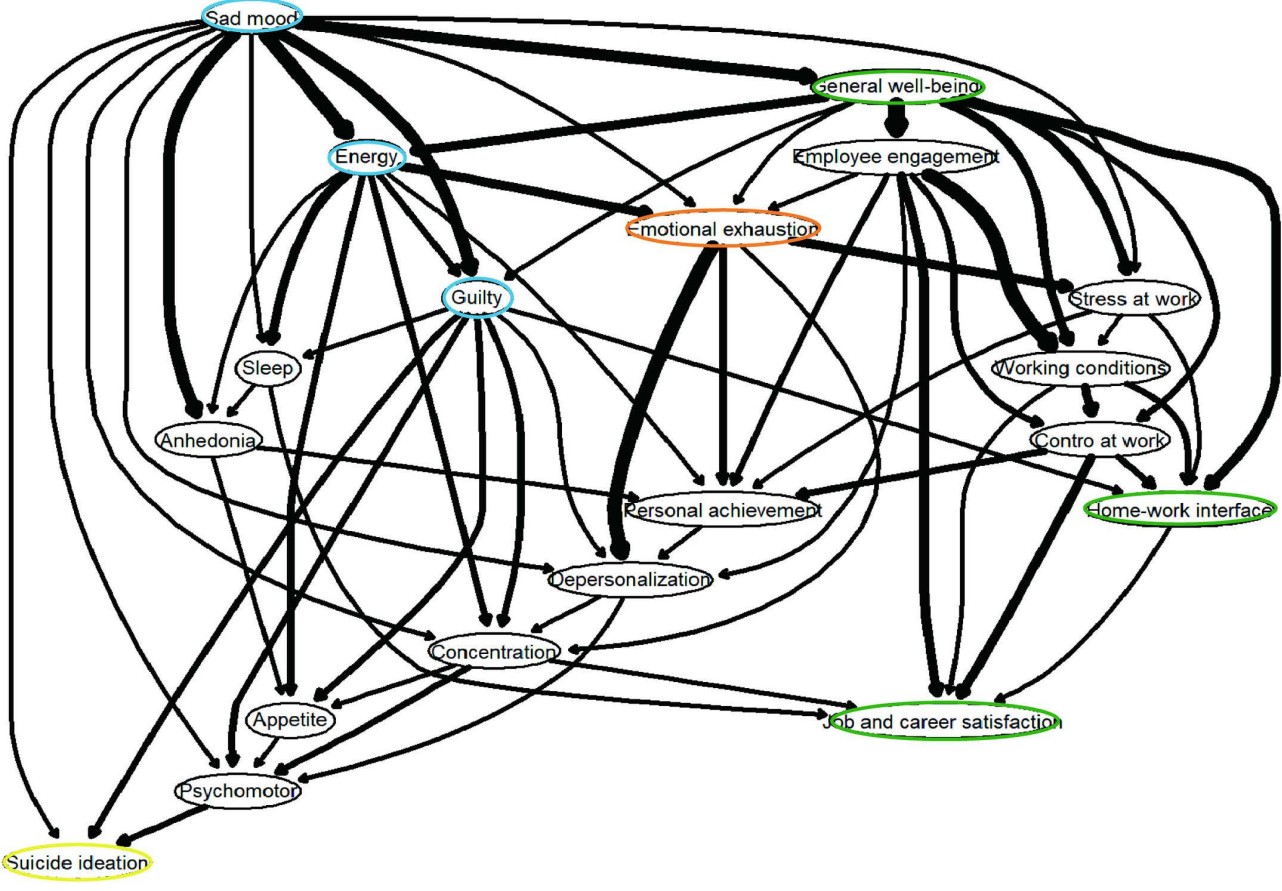

**Fig 3. Directed acyclic graph of the depressive symptoms, burnout, quality of work-related life, and suicidal ideation.**

## Discussion

### Suicidal ideation among psychiatric nurses

The findings from this study show that 11.33% of psychiatric nurses reported experiencing SI in the past two weeks, which is considerably higher than the lifetime prevalence of 3.9% reported in the general Chinese population [33]. This striking disparity underscores the critical role of occupational stressors in exacerbating mental health challenges among psychiatric nurses [34]. The prevalence observed in this study is consistent with recent studies on Chinese nurses, which have reported SI rates ranging from 9.3% to 10.8% [35,36]. Our findings also reinforce evidence that psychiatric nurses are at a particularly high risk for suicide compared with other healthcare professionals. For example, studies have shown that psychiatric nurses are more likely to report SI than their peers are, with prevalence estimates reaching 14.6% for recent SI and up to 39.1% for lifetime SI in some samples [37,38]. The increased prevalence of SI emphasizes the urgent need for targeted mental health support in this professional group.

### Pathway 1: From sad mood, general well-being, and work-home interface to job and career satisfaction

Our study identifies a pathway in which sad mood undermines general well-being and work-home interface, ultimately reducing career satisfaction. While previous research has examined the individual effects of mood, well-being, and career

satisfaction on mental health outcomes, our study is the first to integrate these factors into a cohesive model, highlighting the cascading effects of emotional states on professional satisfaction. Our findings align with previous research underscoring the importance of work-life balance in maintaining career satisfaction [39]. An imbalance between work and personal life has been associated with heightened stress and emotional exhaustion, both of which are negatively affect career satisfaction [40,41]. Consistent with the job demands-resources (JD-R) model, sufficient job resources, such as supportive work environments, opportunities for autonomy, and professional development, can buffer the negative impact of high job demands and help nurses manage their workload depleting personal resources [42,43].

Therefore, improving working conditions and promoting a better work-life balance could help alleviate emotional exhaustion while simultaneously enhancing career satisfaction and overall well-being. Consistent with the JD-R model, targeted interventions that support emotional health, such as structured mental health services, resilience-building strategies, and supervisory support, may yield meaningful improvements in both well-being and career satisfaction. Given the emotionally demanding nature of psychiatric nursing, even modest improvements in these areas are likely to yield not only increased professional fulfillment and quality of life but also broader organizational benefits, including reduced turnover intentions and improved quality of care [43–45].

### Pathway 2: From sad mood, and low energy to emotional exhaustion and suicidal ideation

The second pathway identified in the DAG analysis highlights how depressive symptoms, particularly sad mood, guilt, and low energy, contribute to emotional exhaustion, which in turn increases the risk of SI. This pattern is consistent with previous findings on the role of guilt [46], psychomotor changes [47], sad mood [24], and appetite changes [48] in predicting SI. Within this pathway, emotional exhaustion emerges as a central "bridge" symptom, linking depressive symptoms to work-related stressors and adverse outcomes such as low job satisfaction. This finding supports the Burnout-Depression Continuum Theory [49], which conceptualizes burnout and depression as interconnected conditions, with exhaustion functioning as a pivotal stage in the escalation from occupational stress to depression and SI. By positioning exhaustion both as a consequence of depressive symptoms and a mediator of SI, our model clarifies the cyclical relationship whereby worsening exhaustion fuels further depressive symptoms, ultimately heightening suicide risk [8,49]

These findings underscore the need for dual-level interventions that recognize the continuum between burnout and depression [50]. At the organizational level, strategies aimed at mitigating exhaustion, such as adequate staffing, manageable workloads, and structured recovery opportunities, are critical, as exhaustion may represent an early point along the burnout–depression spectrum. At the individual level, timely screening and intervention for depressive symptoms, alongside support for energy restoration (e.g., sleep hygiene, fatigue management), can help prevent further progression along this continuum. Complementary resources that enhance personal accomplishment, including recognition programs, career development pathways, and peer support, may buffer against the detrimental effects of exhaustion, foster resilience, and potentially interrupt the trajectory from occupational stress to clinical depression.

In addition to confirming prior evidence, this research makes several original contributions. By applying network analysis and Bayesian DAG modeling, we move beyond simple correlations to uncover the causal structures that connect mood, energy, exhaustion, and suicidality. The identification of emotional exhaustion and low energy as "bridge symptoms" offers concrete leverage points for intervention, extending the JD-R model and Burnout–Depression Continuum Theory with new empirical evidence from psychiatric nurses.

Several limitations of this study should be acknowledged. First, this study focused on psychiatric nurses in Western China, which may restrict the generalizability of the findings to other healthcare populations or regions. Second, while the use of DAGs was appropriate for addressing our research objectives, the cross-sectional design of the study prevents us from establishing causality inference. Nevertheless, it enables us to uncover complex interrelationships among the variables, offering valuable insights. Third, participant recruitment through a self-developed digital platform may have introduced some selection bias, although the platform incorporated quality control measures to enhance data reliability. Fourth,

reliance on self-reported measures raises concerns of reporting bias, especially for sensitive issues such as SI, where underreporting due to stigma or fear of repercussions cannot be excluded. Nevertheless, perceptions of emotions, stress, and SI are intrinsically subjective and thus most appropriately assessed through self-reports.

Future studies should employ longitudinal and cross-cultural designs to validate these pathways and examine the long-term effectiveness of both organizational and clinical interventions. Special attention should be given to minimizing response bias in self-reported suicidal ideation, possibly through indirect questioning or mixed-methods approaches. Strengthening the evidence base will not only advance scholarly understanding but also inform practical, evidence-driven strategies to protect psychiatric nurses' mental health and improve the sustainability and quality of mental health care systems.

## Conclusion

In summary, this study identified two key pathways linking depressive symptoms, emotional exhaustion, and job satisfaction to suicidal ideation among psychiatric nurses. The findings emphasize the urgent need to address occupational stressors in this vulnerable workforce through improved working conditions, adequate staffing, and accessible psychological support. By targeting both organizational resources and individual well-being, healthcare institutions can increase job satisfaction, mitigate burnout, and reduce suicide risk, and ultimately benefit both staff and patients.

## Supporting information

**S1 Table. Descriptive statistics of continuous variables (overall sample).** IQR, Interquartile range.
(DOCX)

**S2 Table. Factors associated with suicidal ideation identified by multivariate logistic regression.** OR, odds ratio; 95% CI, 95% confidence interval.
(DOCX)

## Acknowledgments

We are grateful to all the individuals who participated in this study.

## Author contributions

**Conceptualization:** Min Wang, Yushun Yan, Xiao Yang, Xiaohong Ma.

**Data curation:** Wanqiu Yang, Ruini He, Lingdan Zhao, Yikai Dou, Qingqing Xiang.

**Formal analysis:** Min Wang.

**Funding acquisition:** Min Wang, Lingdan Zhao, Xiao Yang, Xiaohong Ma.

**Investigation:** Min Wang, Yushun Yan, Wanqiu Yang, Ruini He, Lingdan Zhao, Yikai Dou, Yuanmei Tao, Xiao Yang, Qingqing Xiang, Xiaohong Ma.

**Project administration:** Qingqing Xiang, Xiaohong Ma.

**Software:** Min Wang.

**Supervision:** Xiaohong Ma.

**Validation:** Min Wang.

**Visualization:** Min Wang.

**Writing – original draft:** Min Wang.

**Writing – review & editing:** Min Wang, Xiao Yang, Qingqing Xiang, Xiaohong Ma.

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
