## [Decision Letter · Decision Letter 0]

19 Aug 2025

PONE-D-25-36942Causal mapping of psychological and occupational risk factors for suicidal ideation in psychiatric nurses using Bayesian networks: A multicenter cross-sectional studyPLOS ONE

Dear Dr. Ma,

Thank you for submitting your manuscript to PLOS ONE. After careful consideration, we feel that it has merit but does not fully meet PLOS ONE’s publication criteria as it currently stands. Therefore, we invite you to submit a revised version of the manuscript that addresses the points raised during the review process.

**ACADEMIC EDITOR:  **

Dear Authors, the manuscript requires minor revisions. Please respond to the reviewers point by point.

Best regards

We look forward to receiving your revised manuscript.

Kind regards,

Omar Enzo Santangelo

Academic Editor

PLOS ONE

Journal Requirements:

4. In the online submission form, you indicated that the data collected and analyzed during the current study are available from the corresponding author (maxiaohong@scu.edu.cn) on reasonable request.

This work was supported by the Ministry of Science and Technology of the People’s Republic of China (grant number 2022ZD0211700), the Key Research and Development Project of Zigong Science and Technology Program (grant number 2023YLWS11), the 135 Project from West China Hospital of Sichuan University (grant numbers 2023HXFH006, 2023HXFH040), and the Postdoctoral Research Fund of West China Hospital, Sichuan University (grant number 2024HXBH135).

6. Please note that funding information should not appear in any section or other areas of your manuscript. We will only publish funding information present in the Funding Statement section of the online submission form. Please remove any funding-related text from the manuscript.

Reviewers' comments:

Reviewer's Responses to Questions

**Comments to the Author**

1. Is the manuscript technically sound, and do the data support the conclusions?

Reviewer #1: Yes

Reviewer #2: Partly

Reviewer #3: Yes

2. Has the statistical analysis been performed appropriately and rigorously? 

Reviewer #1: Yes

Reviewer #2: Yes

Reviewer #3: Yes

3. Have the authors made all data underlying the findings in their manuscript fully available?

Reviewer #1: Yes

Reviewer #2: Yes

Reviewer #3: Yes

4. Is the manuscript presented in an intelligible fashion and written in standard English?

Reviewer #1: No

Reviewer #2: Yes

Reviewer #3: Yes

5. Review Comments to the Author

Reviewer #1: Recommendations for Manuscript ID PONE-D-25-36942 Title: “Causal mapping of psychological and occupational risk factors for suicidal ideation in psychiatric nurses using Bayesian networks: A multicenter cross-sectional study” for the Plos One Journal.

General Comments

From my point of view, it is a very interesting topic and simultaneously it seems that to the best of my knowledge is an empirical research aimed to present that psychiatric nurses represent a high-stress occupational group that experiences elevated levels of suicidal ideation (SI), which necessitates greater attention to their mental health. A total of 1,835 psychiatric nurses participated in the study, completing questionnaires that assessed depressive symptoms (PHQ-9), SI, quality of work-related life (QWL), and occupational burnout. Multivariate logistic regression and phenotypic network analyses, were conducted to identify factors associated with SI and explore the interactions and possible causal directions among depressive symptoms, burnout, QWL, and SI. The findings indicate that 11.33% of psychiatric nurses reported experiencing SI in the past two weeks. The multivariate logistic regression analysis identified emotional exhaustion, depersonalization, personal accomplishment, stress at work, general well-being, and home-work interface were significant as predictors of SI. In the symptom network, the top five symptoms directly connecting SI were psychomotor changes, guilt, sad mood, low energy, and appetite changes. Pathways were observed from sad mood, general well-being, and work-home interface to job and career satisfaction, as well as from sad mood, low energy to emotional exhaustion and SI. The study highlights the importance of targeting sad mood, psychomotor changes, low energy, emotional exhaustion, and work-related stress to reduce the risk of SI among psychiatric nurses.

The paper contains the following sections: Introduction, Methods, Results, Discussion.

However, I find some recommendations:

1. The Manuscript needs careful English proofreading because there are some shortcomings. For instance, the article “the” is sometimes missing in front of nouns, the message in some paragraphs is not clear enough. It looks like the first part was written by one author with a greater command of the English language, and the rest of the paper was written by someone else. The numerous grammar errors made this a difficult paper to read. It was strange to see the authors refer to tables that were not submitted. I was unable to find any supplementary material to the submission, so I think this was truly omitted by the authors. Please read the manuscript carefully.

2. The abstract must contain the main purpose of the paper, the research method used in the research and the main contributions.

3. It would be very useful to add in the "Introduction" section the purpose, objectives and hypothesis of the research. I consider that a weak point of the paper is that the authors did not show the novelty of the paper compared to other works. That is why, I consider that the introduction should specify the novelty of the paper compared to other papers published in this area.

4. Literature review and Conclusion sections cannot be missing from the paper.

5. The research is well based on science and the results are in agreement with the theoretical part.

6. I believe that the authors should also include other indicators from Descriptive Statistics.

7. I think that the literature needs to be improved with other recent works, refers to rural tourism. That is why I recommend the authors to refer to other recent works indexed in Web of Science, Scopus, Emerald and Cambrige Journals. We suggest that the authors cite papers published in Web of Science Journals such as:

a. https://doi.org/10.3390/jrfm14070336

b. https://doi.org/10.1007/s13132-023-01565-6

c. https://doi.org/10.3390/fractalfract8010032

8. In conclusion, the article should be improve. It should also be enhanced with a review of the literature adequate to the subject and a broader interpretation and commentary of the research results.

Reviewer #2: The article does not specify the criteria for inclusiveness and exclusivity of the selection of respondents. A complete description of the respondents is not provided.

The demographic and age distribution is not clear and therefore it is not clear whether the existing sample is sufficiently scientifically reliable and whether it corresponds to the general population.

Reviewer #3: Suicidal ideation (SI) is more common among psychiatric nurses, a high-stress occupational group that requires more attention to their mental health. The study focuses on a significant topic. My observations are given below:

(1) The study title is okay.

(2) The abstract needs revision. The abstract does a good job of summarizing the goal, methodology, and main conclusions of the study, but it is extremely technical and dense, which makes it difficult to read. While practical implications and novelty should be more prominently highlighted, excessive statistical detail could be trimmed. Increased clarity and reader engagement would result from a more focused emphasis on psychological significance.

(3) The introduction also needs minor revision. A thorough and thoroughly referenced background is provided in the introduction, which successfully places the study in the context of psychiatric nurses' risk of suicide. Nevertheless, it is excessively long and has repetitions that mask the main research void. It feels sudden and could be more smoothly integrated to go from general suicide risk to methodological options (network analysis, DAGs). Readability and theoretical coherence would be enhanced by more precise articulation of hypotheses and a greater emphasis on the psychological processes that connect burnout, QWL, and suicidal thoughts.

(4) The method again needs minor amendments. The methods section is thorough, shows exacting statistical and measurement procedures, and reports on the reliability of every instrument. However, generalizability may be limited due to the cross-sectional design, which limits causal inference, and the lack of discussion regarding the rationale behind the exclusive selection of psychiatric nurses from Sichuan Province. Although DAG and network analyses are novel, assumptions like multivariate normality need more convincing evidence. Deeper thought should be given to the selection bias and data validity issues brought up by the hiring process using a self-developed digital platform.

(5) The results need modifications. With its thorough analysis and integration of several statistical techniques, the results section provides deep insights into the connections between SI, burnout, QWL, and depressive symptoms. On the other hand, overly detailed figures and edge weights make it difficult to read and hide important points. Without a clear synthesis, some results are replicated in network and regression analyses. Interpretation would be strengthened by a stronger focus on the practical implications of effect sizes and a more distinct differentiation between statistical significance and clinical relevance. Simplifying would increase clarity without sacrificing the depth of the science.

(6) The discussion needs minor improvements. The discussion is comprehensive, clearly defining two main pathways and connecting findings to pertinent theories and earlier research. Integration of the Burnout-Depression Continuum Theory and the JD-R model is one of its strengths. Nevertheless, some passages are repetitive and excessively detailed, which lessens their impact. Its applicability would be increased by placing more focus on original contributions and useful intervention techniques. Although the limitations section is appropriate, it could more thoroughly address the possibility of response bias in delicate subjects like SI. All in all, a good but unduly long conversation.

6. PLOS authors have the option to publish the peer review history of their article (what does this mean?). If published, this will include your full peer review and any attached files.

Reviewer #1: No

Reviewer #2: **Yes: **Agita Abele

Reviewer #3: **Yes: **Gyanesh Kumar Tiwari

---

## [Author Response · Author response to Decision Letter 1]

29 Aug 2025

1. Please ensure that your manuscript meets PLOS ONE's style requirements, including those for file naming. The PLOS ONE style templates can be found at https://journals.plos.org/plosone/s/file?id=wjVg/PLOSOne_formatting_sample_main_body.pdf and https://journals.plos.org/plosone/s/file?id=ba62/PLOSOne_formatting_sample_title_authors_affiliations.pdf.

Response: We appreciate the editor’s guidance on this matter. We have carefully revised the manuscript to ensure that it meets PLOS ONE’s style requirements, including file naming and formatting, in accordance with the official style templates.

Response: We have revised the manuscript accordingly and deleted the redundant ethics statement from all the sections other than the Methods section, as requested.

Response We thank the editor for the comment regarding data availability. In accordance with the journal’s policy, we have deposited the minimal anonymized data set required to replicate our findings in a stable public repository. The Data Availability statement in the manuscript has also been updated accordingly.

“Data Availability Statement

Data are available from the public data repository OpenICPSR at https://www.openicpsr.org/openicpsr/project/237382/version/V1/view.” (Lines 1415-1416)

4. In the online submission form, you indicated that the data collected and analyzed during the current study are available from the corresponding author (maxiaohong@scu.edu.cn) on reasonable request.

Response: We thank the editor for pointing this out. We have now deposited the minimal anonymized data set underlying the findings of this study in the public data repository OpenICPSR (https://www.openicpsr.org/openicpsr/project/237382/version/V1/view). The Data Availability statement has been updated accordingly.

This work was supported by the Ministry of Science and Technology of the People’s Republic of China (grant number 2022ZD0211700), the Key Research and Development Project of Zigong Science and Technology Program (grant number 2023YLWS11), the 135 Project from West China Hospital of Sichuan University (grant numbers 2023HXFH006, 2023HXFH040), and the Postdoctoral Research Fund of West China Hospital, Sichuan University (grant number 2024HXBH135).

Please state what role the funders took in the study. If the funders had no role, please state: "The funders had no role in study design, data collection and analysis, decision to publish, or preparation of the manuscript. "

Response: We confirm that the financial disclosure statement has been provided correctly and that no further changes are needed. Additionally, as requested, we have included the role of the funder statement in the cover letter as follows:

6. Please note that funding information should not appear in any section or other areas of your manuscript. We will only publish funding information present in the Funding Statement section of the online submission form. Please remove any funding-related text from the manuscript.

Response: We thank the editor for the reminder. We have carefully checked the manuscript and removed all funding-related text. The funding information is now provided only in the Funding Statement section of the online submission form, in accordance with the journal’s requirements.

Response: We appreciate the editor’s helpful reminder. We will carefully evaluate the suggested references and cite them if they are relevant to our study.

Response: We have carefully reviewed the entire reference list to ensure that all entries are complete and correct. No retracted articles are cited in our manuscript. All the references have been double-checked for accuracy, and the updated reference list is included in the revised submission.

Review Comments to the Author

Reviewer #1: Recommendations for Manuscript ID PONE-D-25-36942 Title: “Causal mapping of psychological and occupational risk factors for suicidal ideation in psychiatric nurses using Bayesian networks: A multicenter cross-sectional study” for the Plos One Journal.

General Comments

From my point of view, it is a very interesting topic and simultaneously it seems that to the best of my knowledge is an empirical research aimed to present that psychiatric nurses represent a high-stress occupational group that experiences elevated levels of suicidal ideation (SI), which necessitates greater attention to their mental health. A total of 1,835 psychiatric nurses participated in the study, completing questionnaires that assessed depressive symptoms (PHQ-9), SI, quality of work-related life (QWL), and occupational burnout. Multivariate logistic regression and phenotypic network analyses, were conducted to identify factors associated with SI and explore the interactions and possible causal directions among depressive symptoms, burnout, QWL, and SI. The findings indicate that 11.33% of psychiatric nurses reported experiencing SI in the past two weeks. The multivariate logistic regression analysis identified emotional exhaustion, depersonalization, personal accomplishment, stress at work, general well-being, and home-work interface were significant as predictors of SI. In the symptom network, the top five symptoms directly connecting SI were psychomotor changes, guilt, sad mood, low energy, and appetite changes. Pathways were observed from sad mood, general well-being, and work-home interface to job and career satisfaction, as well as from sad mood, low energy to emotional exhaustion and SI. The study highlights the importance of targeting sad mood, psychomotor changes, low energy, emotional exhaustion, and work-related stress to reduce the risk of SI among psychiatric nurses.

Response: We sincerely thank the reviewer for the encouraging and positive comments on our work.

The paper contains the following sections: Introduction, Methods, Results, Discussion.

However, I find some recommendations:

1. The Manuscript needs careful English proofreading because there are some shortcomings. For instance, the article “the” is sometimes missing in front of nouns, the message in some paragraphs is not clear enough. It looks like the first part was written by one author with a greater command of the English language, and the rest of the paper was written by someone else. The numerous grammar errors made this a difficult paper to read. It was strange to see the authors refer to tables that were not submitted. I was unable to find any supplementary material to the submission, so I think this was truly omitted by the authors. Please read the manuscript carefully.

Response: We sincerely thank the reviewer for the valuable feedback. In the revised manuscript, we have thoroughly proofread the text for grammar, style, and clarity, corrected errors and rephrased ambiguous passages to increase readability and consistency.

With respect to the tables, the main text contains only one table, which has been retained. We have carefully rechecked the revised version to ensure that all the table references are accurate and consistent throughout the manuscript. Furthermore, as suggested, we have added two supplementary tables to the Supporting Information:

“S1 Table. Descriptive statistics of continuous variables (overall sample)

S2 Table. Factors associated with suicidal ideation identified by multivariate logistic regression”

2. The abstract must contain the main purpose of the paper, the research method used in the research and the main contributions.

Response: Following the recommendation, we have revised the Abstract to more clearly articulate the primary purpose of the study, the research methods employed, and the key contributions. The revised Abstract is presented below for your kind consideration:

“Psychiatric nurses represent a high-stress occupational group that experiences elevated levels of suicidal ideation (SI), emphasizing the need for focused mental health interventions. The main purpose of this study was to examine the prevalence of SI among psychiatric nurses and to identify the psychological and occupational factors associated with SI. A total of 1,835 psychiatric nurses completed questionnaires on depressive symptoms (PHQ-9), SI, quality of work-related life (QWL), and burnout. Multivariate logistic regression and phenotypic network analyses were conducted to identify factors associated with SI and the potential pathways linking depressive symptoms, burnout, and QWL to SI. The results indicated that 11.33% of the participants had SI in the past two weeks. Multivariate logistic regression revealed that emotional exhaustion, depersonalization, personal accomplishment, stress at work, general well-being, and the home-work interface were significant predictors of SI. Network analysis further revealed that psychomotor changes, guilt, sad mood, low energy, and appetite changes were the symptoms most directly associated with SI. In addition, sad mood, general well-being, and work-home interface were linked to job and career satisfaction, whereas sad mood and low energy were associated with emotional exhaustion and SI. These findings contribute valuable large-scale evidence on the mental health challenges faced by psychiatric nurses and highlight the importance of addressing mood disturbances, energy loss, and work-related stress in SI prevention efforts for this vulnerable group.” (Lines 24-43)

3. It would be very useful to add in the "Introduction" section the purpose, objectives and hypothesis of the research. I consider that a weak point of the paper is that the authors did not show the novelty of the paper compared to other works. That is why, I consider that the introduction should specify the novelty of the paper compared to other papers published in this area.

Response: We thank the reviewer for the insightful comment. Following this suggestion, we have revised the Introduction to include the purpose, objectives, and hypothesis of the study. The revised manuscript is as follows:

“The purpose of this study was to investigate the prevalence of SI among psychiatric nurses and to examine how depressive symptoms, burnout, and QWL interact to influence SI. We hypothesized that higher levels of burnout and depressive symptoms, coupled with lower QWL, would be associated with an increased risk of SI. To test these hypotheses, we applied multivariate regression to identify significant predictors and employed network analysis together with Bayesian causal models to capture the complex interrelationships among these factors. This integrative methodological approach extends previous research by moving beyond correlation-based analyses to infer potential causal pathways, thereby offering novel insights into central and bridge symptoms that may serve as effective intervention targets to reduce SI and improve the mental health of psychiatric nurses.” (Lines 108-118)

In addition, we have clarified the novelty of our research by emphasizing how our work integrates depressive symptoms, burnout, and QWL within a network and Bayesian framework, which differs from prior studies that examined these factors separately. The revised manuscript is as follows:

“Despite the recognized importance of burnout, QWL, and depressive symptoms in predicting suicide risk among healthcare professionals, most previous studies have examined these factors in isolation. This fragmented approach obscures the complex and dynamic interactions among them and limits the understanding of their combined impact on the SI. Addressing this gap requires an integrative approach that can capture both the complexity of symptom interactions and the potential causal pathways underlying them.

Network analysis provides such a framework. Unlike traditional methods that typically focus on linear relationships, network analysis visualizes and quantifies the relationships among how symptoms cluster and interact within a broader structure[18]. This approach has been particularly useful in mental health research, offering insights into the interplay of multiple symptoms and hi

---

## [Decision Letter · Decision Letter 1]

8 Sep 2025

Causal mapping of psychological and occupational risk factors for suicidal ideation in psychiatric nurses using Bayesian networks: A multicenter cross-sectional study

PONE-D-25-36942R1

Dear Dr. Ma,

We’re pleased to inform you that your manuscript has been judged scientifically suitable for publication and will be formally accepted for publication once it meets all outstanding technical requirements.

Kind regards,

Omar Enzo Santangelo

Academic Editor

PLOS ONE

Additional Editor Comments (optional):

Reviewer #4:

Reviewer #5:

Reviewers' comments:

Reviewer's Responses to Questions

**Comments to the Author**

1. If the authors have adequately addressed your comments raised in a previous round of review and you feel that this manuscript is now acceptable for publication, you may indicate that here to bypass the “Comments to the Author” section, enter your conflict of interest statement in the “Confidential to Editor” section, and submit your "Accept" recommendation.

Reviewer #4: (No Response)

Reviewer #5: All comments have been addressed

2. Is the manuscript technically sound, and do the data support the conclusions?

Reviewer #4: Yes

Reviewer #5: Yes

3. Has the statistical analysis been performed appropriately and rigorously? 

Reviewer #4: Yes

Reviewer #5: Yes

4. Have the authors made all data underlying the findings in their manuscript fully available?

Reviewer #4: Yes

Reviewer #5: Yes

5. Is the manuscript presented in an intelligible fashion and written in standard English?

Reviewer #4: Yes

Reviewer #5: Yes

6. Review Comments to the Author

Reviewer #4: Thank you for submitting the revised version of your manuscript. I have carefully reviewed the changes made in response to the previous comments, and I am pleased to see that you have addressed all of the concerns raised in a thoughtful and comprehensive manner. The manuscript has improved significantly in clarity, structure, and scientific rigor. The revisions enhance the overall quality of the work and contribute positively to the field. I believe the current version meets the standards required for publication. Congratulations on a job well done, and I wish you the very best with the publication.

Reviewer #5: Dear Author(s),

The manuscript entitled “Causal mapping of psychological and occupational risk factors for suicidal ideation in psychiatric nurses using Bayesian networks: A multicenter cross-sectional study” addresses an important and timely topic with high relevance to both psychiatric nursing and mental health research. The study is based on a large multicenter sample of psychiatric nurses, applying validated instruments and rigorous statistical approaches, including multivariate logistic regression, network analysis, and Bayesian modeling. This methodological combination allows the authors to move beyond simple associations and to explore potential causal pathways, which represents a significant strength and an original contribution to the literature.

The paper is now clearly written, logically structured, and in line with the journal’s formatting requirements. The abstract and introduction effectively present the purpose, objectives, and novelty of the research. The methods section is detailed and transparent, with inclusion/exclusion criteria and quality control procedures clearly described. The results are presented in a balanced way, with appropriate attention to both statistical and clinical significance. The discussion integrates the findings with theoretical frameworks such as the JD-R model and the Burnout–Depression Continuum, while also highlighting practical implications for interventions targeting mood disturbances, energy loss, and occupational stress among psychiatric nurses.

Overall, this is a well-executed and important study that provides novel insights and practical implications for the prevention of suicidal ideation in a vulnerable occupational group. I recommend acceptance of this manuscript for publication in PLOS ONE.

Sincerely,

Reviewer

7. PLOS authors have the option to publish the peer review history of their article (what does this mean?). If published, this will include your full peer review and any attached files.

Reviewer #4: No

Reviewer #5: No

---

## [Editor Report · Acceptance letter]

PONE-D-25-36942R1

PLOS ONE

Dear Dr. Ma,

I'm pleased to inform you that your manuscript has been deemed suitable for publication in PLOS ONE. Congratulations! Your manuscript is now being handed over to our production team.

Kind regards,

on behalf of

Dr. Omar Enzo Santangelo

Academic Editor

PLOS ONE